# Digital boundaries: A study on WeChat parent-child relationships among Chinese college students and their association with family factors

**Zhao Feng**[1], **Chun-Mei Hu**[2]*, **Ling-Ling He**[3]

**1** School of Literature, Journalism and Communication, Xihua University, Chengdu city, Sichuan province, China, **2** Mental Health Education and Counseling Center, Chongqing University of Arts and Sciences, Chongqing city, China, **3** School of Intelligent Manufacturing Engineering, Chongqing University of Arts and Sciences, Chongqing city, China

* zeefaye@163.com

## Abstract

### Objective

This study aims to investigate the current state of WeChat parent-child relationships among college students and their association with family factors.

### Methods

Using convenient sampling, 6159 students,age from 18 to 25,the participants were selected from colleges of Chengdu to complete a questionnaire survey on response and attitudes towards parents' WeChat friend requests, Family APGAR Index,Parental Management Scale,Family Conflict Scale,were used as measurement tools.

### Results

Responses to parental WeChat friend requests showed significant positive correlations with higher levels of family function satisfaction ($r = 0.10$, $P<0.01$) and parental management ($r = 0.11$, $P<0.01$), and a significant negative correlation with family conflict ($r = -0.03$, $P<0.05$).

### Conclusion

This study reveals that Chinese college students generally accept their parents' WeChat friend requests, influenced by family dynamics and traditional values. Positive parental management, family harmony, and good marital relationships enhance students' willingness to connect with their parents on WeChat, indicating the complex interplay between digital interactions and traditional family concepts.This underscores the importance for parents to adopt modern and scientific parenting approaches, fostering a harmonious and nurturing family atmosphere rooted in trust and mutual respect for their children.

**Data Availability Statement:** The raw data supporting the findings of this study are available at DOI:10.6084/m9.figshare.25139894.

**Funding:** The article was supported by the following funds 1.The Chongqing Higher Education Teaching Reform Research Project "Exploration and Practice of the Construction of College Teachers' Psychological Education Capacity" (233388) 2.The Chongqing Municipal Education Commission Humanities and Social Sciences Research Project "Research on Enhancing the Targeted and Effective Psychological Health Education for College Students" (23SKSZ051).

**Competing interests:** The authors have declared that no competing interests exist.

## Introduction

According to data released by Tencent in August 2023, the number of Wechat monthly active accounts has reached 1.327 billion. With the large-scale coverage of the population, WeChat has been called "the first national social app" in China. The social relationship of WeChat platform has also broken through the original concept of "virtualization" of the Internet, and almost relocated the real social network of Chinese people to this APP, and the ensuing problem of interpersonal relationship has gradually become a topic for the relationship of scientific researchers.

Wu Zhihong, one of China's leading psychological counselling experts, once said: The parent-child relationship exists ultimately for the purpose of separation. For Chinese adolescents, the university stage is the first step to complete the "separation" of the parent-child relationship, when most adolescents will leave their families for the first time and become self-managed. In the Chinese cultural context, adolescents at this stage "seek independence and privacy, but are also dependent on their parents," and Chinese parents at this stage "are used to monitoring their children's behaviour and psychology, but must also accept the fact that their children are growing up and detaching from their families of origin" [1].

WeChat, as a platform for starting, developing, maintaining and ending various social relationships, is an important tool for college students who have left their families to maintain contact with their parents, and plays a complex role in adolescents' privacy management and response to self-disclosure. The complexity of social relationships on the WeChat platform also poses a challenge to the self-disclosure mechanism of college students, forcing them to strike a dynamic balance between content sharing, self-disclosure, privacy boundary maintenance, and between the pursuit of self-awareness and the maintenance of family ties.

### Parent-child relationship and adolescents' self-disclosure on social media

Social media has become an important medium for communication within contemporary families, serving as a crucial tool for parent-child interaction. However, a typical characteristic of adolescents' internet behavior is their efforts to prevent their online privacy from being exposed to their parents rather than to strangers [2]. Self-disclosure is the first driver of social media use, and users gain social support and social capital by sharing and exchanging personal information on social media platforms. Therefore, we take the establishment of WeChat relationship, especially the visibility of WeChat moments (WeChat Moments is a feature on the WeChat social networking platform that allows users to post text, pictures, videos and other content in their personal space to share life updates, opinions and feelings with their WeChat friends), as an important factor in examining the response of college students to selectively self-disclosure. Since social media emerged earlier in the West, Western researchers paid attention to this issue earlier, Petronio found that The privacy management of social media platforms predominantly revolves around self-disclosure and boundary management [3], and the personal privacy management strategies of social media users are products of cultural factors [4], Family factors also play an important role in influencing children's attitudes towards their parents' social media friends, as the family's privacy orientation fosters the construction of communication habits and privacy boundaries with family member [3], children's response to add their parents as social media friends is one of the conditions to consider their attitudes to self-disclosure and privacy boundary management; Parent-child relationship, family satisfaction, as well as the different types of families, have an important impact on adolescents' self-disclosure practices and values [5]. Families can be the first shaping force in influencing young people's attitudes and approach to information disclosure. The huge scale of WeChat's users will provide a valuable reference for the study of the privacy boundaries between adolescents

and their parents in the age of social media, and the dialectical impact of parent-child relationships; and China's long historical and cultural background has been extremely permeable to Chinese values, family relationship patterns, and notions of privacy, but at the same time the privacy boundaries of offspring leaving their families will change, and a more complex set of rules for the practice of self-disclosure will be formed under the combined influence of the family of origin and the subsequent environment [6].

## Parent-child relationships in the context of traditional Chinese culture: The inequality of privacy power

Chinese norms of "human-relationship" begin within the family [7], Confucius, the spiritual leader of Confucianism, proposed a set of internal ethical systems for the Chinese family, in which "father's kindness and son's filial piety" are the first internal ethical rules of the family, which emphasizes two elements of family ethics in traditional Chinese culture: the first point is that parents are the authority of the family, weakening the importance of the principle of "paternal kindness" (i.e., parents must care for their children) while emphasizing "the father is the principle of the son". Therefore, the second point, the obedience of children to their parents as emphasized in "Zi Xiao" (children's filial piety, harmony and absolute obedience). "Filial piety" unilaterally emphasises the obligation of children to their parents, and it has been expanded into the term "filial piety and obedience", which is used as a key factor in measuring the obligation of children to their parents and the moral standard of the children; in the term "filial piety and obedience" the word "obedience" is an important lexical component, and from this derives the moral concepts of "parents' orders cannot be disobeyed" in Chinese traditional society.

The above discussion implies that family relationships in traditional Chinese culture are hierarchical, which leads to the inevitable inequality of family relationships under the traditional Chinese concept of the family, and the right to privacy of children is far less than that of their parents, parents being required to know all the details of the children's lives, for example, in the case of families in ancient China, the parents were required to be informed of the children's success or failure to have sexual relations on their wedding night.

Fei (1983) points out that the pattern of intergenerational relations in the Chinese family is a "feedback pattern" This pattern is different from the unidirectional transmission pattern of the Western family, indicating that the degree of closeness of family relations in China is much higher than in the West [8]. Unlike Western culture, which emphasizes the "self", traditional Chinese culture places more emphasis on the concept of the "family-centered self" [9]. This is a collectivist value that permeates the Chinese management of privacy, with the classification of privacy as "solitude" belonging to the individual and "intimacy" belonging to the group [10], while privacy in traditional Chinese culture belongs more to the latter, defining the boundaries of privacy in terms of the family as a unit, which is a kind of concept of privacy under the collectivist view, in this conception of privacy, the family is an indivisible whole, and family members have an obligation to protect the security of privacy within the family from the outside world, but for within the family, the boundaries of privacy between family members are blurred and not valued [11].

## Digital manifestation of parent-child relationships in China: The restructuring of family structures and power dynamics through WeChat

With the emergence of social networks, the traditional structure of family relationships in China has been broken and rebuilt. The emergence of the Internet has shaken the absolute authority of parents in traditional Chinese family relations, and family power relations have

changed from one-way inter-generational authority to two-way authority [12], China's youn-ger generation, called digital natives, is generally more capable of using the Internet than their parents' generation, so in the digital world, parents even need to learn from their children the skills to integrate into the Internet world.

For Chinese college students who have left their homes, WeChat serves as a vital medium for sustaining communication with their parents. Hence, the parent-child relationship within the framework of the WeChat platform can be termed as the "WeChat parent-child relation-ship," which also functions as a crucial reference point for examining the dynamics of Chinese parent-child interactions within the context of new media.

Researchers have placed significant emphasis on the role of WeChat in Chinese families and parent-child relationships. Various studies have affirmed WeChat's crucial role as a tool for facilitating internal family communication [1, 12, 13], leading to an increase in the fre-quency of family interactions, and making family communication a part of daily life [14]. The number of middle-aged and elderly Internet users in China is also steadily rising, accounting for 28% of the total user base [15] (CNNIC,2023). WeChat has emerged as a social media plat-form with high user stickiness among middle-aged and elderly users, thereby becoming an essential tool for intergenerational communication [14].

However, it has also been pointed out that Chinese young people have passive resistance and avoidance reactions to information shared by their parents on WeChat [16]. In addition to intergenerational differences in preferred content, the low media literacy of the parents' generation leads to the low quality of the content shared, which is also the reason for the chil-dren's resistance and avoidance of receiving content shared by their parents[14].

Research has indicated that the more harmonious the intergenerational relationship, the more intimate the information interaction behavior within the family [17]. The study results demonstrate that as children enter university, mothers become the primary agents in forward-ing online information to their children via WeChat. However, Duan [14] describe the WeChat information sharing from parents to children as a "continuation of the Pre-figurative culture." In this context, parents habitually send information to their children from the per-spective of educators and guides, representing a top-down model. This contrasts with the emerging trend of post-figurative culture, which has gradually gained prominence——this cul-ture points out that under the catalyst of the technological wave represented by the Internet, young people, as digital natives, have become the power holders of the new world culture, thus providing a mode of cultural feedback to the previous generation [18]. The children's passive resistance to their parents' WeChat sharing behavior reflects inter-generational differences in evaluating social media as a tool for parent-child communication—similar findings emerge in studies related to Facebook: parents view Facebook as an effective tool for enhancing parent-child communication, while children perceive it as diminishing the quality of such communi-cation [19].

Based on the above, "WeChat parent-child relationships" can be considered a branch of research on parent-child relationships, representing a digital manifestation of Chinese parent-child relationships. We will use university students' privacy management and information dis-closure methods on WeChat regarding their parents as key factors to assess the quality of WeChat parent-child relationships. We refer to Communication Privacy Management (CPM) theory [3]. The Communication Privacy Management theory (CPM) argues that individuals believe they have ownership and control of their privacy and have the right to determine the boundaries of privacy management. However, when people express themselves in social media (in the case of WeChat, mainly referring to the content posted in Moments), the publishers tacitly regard the people who can see the content as the co-owner of the personal disclosure (privacy) information,For WeChat users, setting boundaries on the content of their WeChat

Moments determines the ownership of their privacy, enabling privacy control and mitigating privacy turbulence. Meanwhile, when disclosure occurs, it signifies an extension of the privacy boundary, transitioning from personal ownership of privacy rights to collective ownership.

Thus, the Responses and Attitudes of College Students toward parental WeChat friend requests can be regarded as the foundation for college students to establish privacy ownership and exercise privacy control. At the same time, it is only after college students add their parents as friends on WeChat and grant them access to view their Moments that true co-ownership is established and privacy boundaries are extended.

On the other hand, when college students regard their parents as targets for self-disclosure, the manner in which they disclose themselves reflects the level of intimacy in the parent-child relationship. In contrast to the unilateral information sharing approach of Chinese parents, Chinese youth deliberately conceal a portion of their social media accounts from their parents; some define their online social interactions as a "space limited to friends, inaccessible to parents," while others manage their personal content through "privacy settings" and "group management." Overall, some researchers argue that the younger generation in China employs various strategies for "digital distancing," attempting to "escape" from their parents to avoid unnecessary trouble and conflicts [20–22], Prevent or terminate the occurrence of privacy turbulence.

As mentioned above,under the influence of the "family-centered self" concept, whether or not a child can be completely honest with their parents is considered a significant criterion for determining the level of filial piety in young Chinese individuals. However, with the strengthening of young people's sense of "self" in contemporary Chinese society, the gradual dissolution of absolute parental authority in the traditional Chinese family, and the emergence of the post-figurative culture, China's "young people feel empowered by the different boundaries offered by the communication that happens online" [23]. Thus, after adding their parents as friends on WeChat, college students may also adjust their privacy management strategies by blocking them from viewing WeChat Moments,prevent or terminate the occurrence of privacy turbulence [3].

In summary, by integrating CPM theory and the features of WeChat, We establish three dimensions for measuring the 'WeChat parent-child relationship' among Chinese college students: their responses to parental WeChat friend requests, their attitudes towards these requests, and whether they block their parents from viewing their WeChat Moments. Currently, research on WeChat parent-child relationships lacks empirical investigation, and this study aims to address that gap. We hypothesize that even though the self-awareness of contemporary Chinese youth is on the rise, the traditional Chinese family values centered on the "filial piety" culturer will still influence information exchange and privacy management practices within Chinese families.

Therefore, we propose Research Hypothesis 1 (H1):

## Chinese college students maintain a positive WeChat parent-child relationship

Family Factors Influencing the WeChat Parent-Child Relationship Among College Students

The family is the most direct influencing factor in the development of college students. Research indicates that family function, management style, and environment significantly impact parent-child relationships. For instance, a well-functioning family can enhance intimacy and adaptability between parents and children, foster healthy parent-child attachment, and effectively reduce adolescents' dependence on smartphones [24].

Dysfunctional family dynamics can negatively impact adolescent mental health. For example, issues such as parental divorce, parental incarceration, parental substance abuse, and

parental mental illness leading to family dysfunction can have extremely detrimental effects on parent-child relationships and result in poor mental health outcomes for children or adolescents within the family [25–27].

Parents, as the primary caregivers of adolescents, influence the parent-child interaction process through their management styles, capabilities, and other aspects of parenting. These factors can affect the children's emotions and behavior, thereby impacting the parent-child relationship [28, 29].

Numerous studies have also demonstrated that parental conflict not only undermines adolescents' self-esteem and contributes to psychological problems but also disrupts the family environment. This, in turn, damages the parent-child attachment and reduces the quality of the parent-child relationship [28, 30, 31].

Based on the aforementioned impact of family factors on parent-child relationships, we hypothesize that these factors will also influence WeChat parent-child relationships. We identify family function satisfaction, parental management, and family conflict as research variables to explore their impact on college students' WeChat parent-child relationships. Thus, we propose the following hypotheses:

Hypothesis 2 (H2): Family function satisfaction will positively influence college students' WeChat parent-child relationships.

Hypothesis 3 (H3): Parental management will positively influence college students' WeChat parent-child relationships.

Hypothesis 4 (H4): Family conflict will negatively influence college students' WeChat parent-child relationships.

## Method

### Research design

This study employs a cross-sectional design. A questionnaire survey is used, which consists of two parts. The first part covers demographic variables, including age, gender, grade level, hometown, major, father's education level, mother's education level, and parental marital status. The second part includes the survey instruments: a questionnaire on response and attitudes towards parents' WeChat friend requests, a family function satisfaction scale, a parental management scale, and a family conflict scale. Chi-square test was employed to assess the relationship between categorical variables.

### Participants

The recruitment period for this study spanned from March 2023 to October 2023. During this time, we distributed questionnaires to prospective participants. Convenience sampling was employed to select 6159 freshmen to junior students to participate in the questionnaire survey at colleges in Chengdu, which is a representative core city in western China. Chengdu, as the central city in western China, attracts students from across the country, making its colleges a microcosm of diverse regional backgrounds. Therefore, we considered Chengdu a representative research location for exploring the attitudes and behaviors of Chinese college students. The age range is 18–25 years, the mean age is (18.96+ 1.01)years. Specifically, there are 2,143 individuals aged 18 (34.8%), 2,459 individuals aged 19 (39.9%), and 1,557 individuals aged 20–25 (25.3%) (since the number of individuals aged 21–25 is relatively small, they have been combined into the 20–25 age group). Students in their fourth year of college in China will leave school and go to different regions to participate in internships and therefore did not participate in this survey. A total of 6,159 valid questionnaires were collected, with a validity rate of 94.75%. Among them, 2,570 were male students, accounting for 41.7%, and 3,589 were

female students, accounting for 58.3%. In terms of academic year, there were 2,190 first-year students (35.6%), 3,075 second-year students (49.9%) and 894 third-year students (14.5%). In terms of place of residence, 5,333 students were from rural areas (86.6%) and 826 students were from urban areas (13.4%). In terms of academic subjects, 3,227 (52.4%) were from liberal arts, 2,057 (33.4%) from the sciences, 496 (8.1%) from physical education and 379 (6.2%) from fine arts. With regard to the educational level of the fathers, 5,656 fathers had a high school education or less (91.8%), and 503 fathers had an associate's degree or higher (8.2%). For mothers, 5,770 had a high school education or less (93.7%), and 389 had an associate's degree or higher (6.3%). Regarding the marital status of the parents, 5,099 were in good marital status (82.8%) and 1,060 were in poor marital status (divorced or separated) (17.2%). The mean age was (18.96 ± 1.01) years. All participants signed an informed consent form before participating in the survey.

## Procedure

The surveyors contacted the counsellors to inform them of the content of the survey and the precautions to be taken, and to determine the time and place of the survey; the counsellors organized the students to go to the classroom to to collectively complete the questionnaires under the supervision of teachers. Investigators on-site to explain the content and purpose of the survey, emphasize the anonymity of the survey and the confidentiality of the results, etc., distributed the survey informed consent form and questionnaire, and asked students to voluntarily sign the informed consent form before completing the questionnaire. After collection, the investigators sorted out the questionnaires, discarded any invalid ones, and proceeded with data entry and analysis.The ethical requirements were met and consent was given by the Ethic Committee of the author's affiliation to conduct the research, and the Ethical review document are available upon request.

## Measures

**Parents' WeChat friend status questionnaire.**   Based on Mullen and Hamilton's investigation of adolescents' response to parental friend requests on social networks [32], where the social network was modified as "WeChat", with 4 questions, 2 questions investigating "Is your father/mother your WeChat friend?" with "No" scoring "0" and "Yes" scoring "1"; and 2 questions on "If your father/mother is your WeChat friend, will you block him/her?" Blocking refers to setting WeChat Moments to be invisible to father/mother", "No"scoring "0 ", "Yes" scoring"1 ". According to the participants' answers, those who added both their fathers and mothers as WeChat friends were categorized as "parents' WeChat friends", and those who blocked both their fathers and mothers from WeChat were categorized as "blocking parents' on WeChat".

**Attitude questionnaire regarding parents' WeChat friends requests.**   Referring to the questions in Mullen's [32] survey on adolescents' attitudes toward their parents' becoming social network friends, the social network was modified to "WeChat", with four questions, sample items include "How do you feel about your father/mother adding you as a WeChat friend?" "How do you feel about using WeChat as a tool to communicate with your father/mother?" The answers were scored on a 5-point scale, from "1" for "very bad " to "5" for "very good". ". The higher the total score, the more positive the attitude towards parents' becoming WeChat friends. The consistency coefficient of this questionnaire in this study was 0.913.

**Family APGAR index.**   Developed by Smilkstein [33], this 5-item instrument measures participants' own subjective ratings of family function satisfaction on a 3-point scale, with "0" representing "hardly ever", "1" representing "sometimes", and "2" representing "often". Sample

statements include "I get satisfactory help from my family when I have problems" "I am satisfied with the way my family discusses things with me and shares problems" "My family is accepting and supportive when I want to take up new activities or developments" etc. Higher total scores indicate better family functioning satisfaction. and more caring families. The consistency coefficient of this questionnaire in this study was 0.911.

**Parental management scale.** Developed by Chen et al. [34] Participants were asked to evaluate their fathers' and mothers' management of themselves in five areas (e.g., their parents' knowledge of their daily lives, their friends, and the activities they do with them, etc.), with a total of 10 questions (5 questions for each parent) on a 4-point scale ranging from"1 point" represents "strongly disagree" to "4 points" represents "strongly agree", sample statements include "My father/mother knows which friends I have", "My father/mother knows what I am doing", "My father/mother knows where I am if I am not at home". The higher the total score, the better the participants perceive their parents' management. The consistency coefficients of this scale in this study were 0. 939 respectively.

**Family conflict scale.** Developed by Espelage et al. [35] (2013) to measure participants' perceptions of hostile conflict situations between family members, the 3-item instrument is scored on a 5-point scale ranging from "1" for "never" to "5" for "always". Sample statements include "In your family, the frequency of yelling, quarrelling, and arguing between your parents is" How often do your parents lose their temper or have emotional breakdowns for no apparent reason" "How often do physical confrontations (e.g., pushing, shoving, throwing things) occur between parents in your household" Higher total scores indicate a greater perceived level of conflict hostility between family members. The coefficient of agreement for this scale in this study was 0.852.

## Statistical analysis

SPSS 21.0 was employed to enter and analyse the data. The count data were described by frequency and percentage, and the measurement data were described by (M±SD). 2-test, independent samples t-test and ANOVA were used to compare the differences in WeChat parent-child relationship among college students with different age,gender,grade,hometown,parents' education level, and parents'marital statues, and correlation analysis was conducted to understand the correlation between college students' WeChat parent-child relationship and family function, parental management, and family conflict, multifactor Logistic regression analysis and stepwise linear hierarchical regression were used to analyse the effects of family caring, parental management, and family conflict on WeChat parent-child relationship. The Harman single-factor method was used to test whether there was any common method bias in the research variables, and the results showed that the variance explained rate of the 1st common factor was 19.24%, which was smaller than the critical value of 40%, indicating that the data in this study did not have any bias in the results caused by similar research methods. The difference was considered statistically significant at P<0.05.

## Result

### Basic information on parental WeChat friend request and moments blocking

Out of the participants, 5937 individuals (96.4%) had their fathers as WeChat friends, while 222 (3.6%) did not; 787 participants (12.8%) had blocked their fathers on WeChat Moments, while 5372 (87.2%) had not.5873 individuals(95.4%)had their mothers as WeChat friends, while 286 (4.6%)did not; 686 participants(11.1%)had blocked their mother's as friends on

WeChat Moments. Both parents are friends on WeChat with 5696 individuals (92.5%), only one parent is friends with 418 individuals (6.8%), and neither parent is friends with both parents with 45 individuals (0.7%). Both parents have not been blocked by 5266 individuals (85.5%), only one parent has been blocked by 213 individuals (5.1%), and both parents have been blocked by 580 individuals (9.4%).

## Parental friend request and moments blocking based on demographic variables

Chi-square test results indicate statistically significant differences in the proportion of college students whose parents are WeChat friends across different grade levels, parental education levels, and marital statuses. (p-value <0.05), with the proportion of first-year students (93.8%) being significantly higher than that of second and third-year students (91.7%, 91.9%), and that the proportion of college students whose parents' education is college degree or above (95.0%) is significantly higher than that of those whose parents' education is high school degree or below (92.3%).), those with good parental marriages (96.2%) were significantly higher than those with poor parental marriages (74.4%); The proportions of college students whose parents are WeChat friends did not show statistically significant differences across different genders, majors, hometowns, and maternal education levels (all p-values > 0.05). See Table 1.

There are significant differences in the proportion of college students blocking their parents on WeChat Moments among different genders, majors, and parental marital status (P values all <0.001). Females (11.7%) are significantly higher than males (6.3%), while students majoring in physical education (12.5%) are notably higher than those in liberal arts (10.5%), science (7.8%), and fine arts (5.3%). Those with parents in poor marital status (13.1%) are significantly higher than those with parents in good marital status (8.6%). See Table 1.

## Attitudes of college students toward parental WeChat friend request across different demographic variables

The score of college students' attitudes towards parental WeChat friend requests was (4.28 ±0.80), which was higher than the median value of 3, indicating that the overall attitudes towards parents becoming WeChat friends were positive.

The results of independent samples t-test and ANOVA showed that the differences in attitude scores towards parents becoming WeChat friends among college students of different genders, grades, majors, parent's education, and parent's marriage were statistically significant, with male students' scores significantly higher than female students', scores of first and third years significantly higher than those of second years, art students significantly lower than those of liberal arts, science, and physical education students, students with parent's education of college degree and above significantly higher than those with parent's education of high school and below, and students with good parental marriages were significantly higher than students with poor parental marriages (all p-values <0.05). There was no statistically significant difference in the attitude scores of students from different hometowns towards parents becoming WeChat friends (P-values > 0.05). See Table 2.

## Correlation analysis between response to parental WeChat friend request and family factors

College students' family APGAR score was (6.16±2.76), parental management score was (26.41±7.09), and family conflict score was (6.03±2.36). The results of correlation analysis showed that responses to parental WeChat friend requests were significantly positively

**Table 1. Differences in parental WeChat friends and blocking parents on WeChat moments among college students with different characteristics.**

| Demographic variables | | Number of people | Parents as WeChat Friends | | Blocking Parents on WeChat Moments | |
|---|---|---|---|---|---|---|
| | | | No($n$%) | Yes($n$%) | No($n$%) | Yes($n$%) |
| Age | 18 yeas old | 2143 | 158(7.4) | 1985(92.6) | 1917(89.5) | 226(10.5) |
| | 19 years old | 2459 | 187(7.6) | 2272(92.4) | 2240(91.1) | 219(8.9) |
| | 20–25 years old | 1557 | 118(7.6) | 1439(92.4) | 1422(91.3) | 135(8.7) |
| $\chi^2$ | | | 0.100 | | 4.972 | |
| P Value | | | 0.951 | | 0.083 | |
| Gender | Male | 2570 | 187(7.3) | 2383(92.7) | 2409(93.7) | 161(6.3) |
| | Female | 3589 | 276(7.7) | 3313(92.3) | 3170(88.3) | 419(11.7) |
| $\chi^2$ | | | 0.369 | | 51.383*** | |
| P Value | | | 0.544 | | 0.000 | |
| Grade | 1 | 2190 | 136(6.2) | 2054(93.8) | 1975(90.2) | 215(9.8) |
| | 2 | 3075 | 255(8.3) | 2820(91.7) | 2798(91.0) | 277(9.0) |
| | 3 | 894 | 72(8.1) | 822(91.9) | 806(90.2) | 88(9.8) |
| $\chi^2$ | | | 8.412 | | 1.205 | |
| P Value | | | 0.015* | | 0.548 | |
| Major | Liberal Arts | 3227 | 254(7.9) | 2973(92.1) | 2889(89.5) | 338(10.5) |
| | Science | 2057 | 146(7.1) | 1911(92.9) | 1897(92.2) | 160(7.8) |
| | Physical Education | 496 | 41(8.3) | 455(91.7) | 434(87.5) | 62(12.5) |
| | Fine Arts | 379 | 22(5.8) | 357(94.2) | 359(94.7) | 20(5.3) |
| $\chi^2$ | | | 3.101 | | 23.844 | |
| P Vale | | | 0.376 | | 0.000*** | |
| Hometown | Rural | 5333 | 409(7.7) | 4924(92.3) | 4839(90.7) | 494(9.3) |
| | Urban | 826 | 54(6.5) | 772(93.5) | 740(89.6) | 86(10.4) |
| $\chi^2$ | | | 1.318 | | 1.484 | |
| P Value | | | 0.251 | | 0.223 | |
| Father Education Level | ≤High-school | 5656 | 438(7.7) | 5218(92.3) | 5130(90.7) | 526(9.3) |
| | ≥College | 503 | 25(5.0) | 478(95.0) | 449(89.3) | 54(10.7) |
| $\chi^2$ | | | 5.112 | | 1.116 | |
| P Value | | | 0.024* | | 0.291 | |
| Mother Education Level | ≤High-school | 5770 | 441(7.6) | 5329(92.4) | 5230(90.6) | 540(9.4) |
| | ≥College | 389 | 22(5.7) | 367(94.3) | 349(89.7) | 40(10.3) |
| $\chi^2$ | | | 2.071 | | 0.365 | |
| P Value | | | 0.150 | | 0.546 | |
| Parents' Marital Status | Poor | 1060 | 271(25.6) | 789(74.4) | 921(86.9) | 139(13.1) |
| | Good | 5099 | 192(3.8) | 4907(96.2) | 4658(91.4) | 441(8.6) |
| $\chi^2$ | | | 599.911 | | 20.505 | |
| P Value | | | 0.000*** | | 0.000*** | |

Note:

* indicates P < 0.05

** indicates P < 0.01

*** indicates P < 0.001.

correlated with higher family function satisfaction (r = 0.10, P<0.01) and parental management (r = 0.11, P<0.01), and significantly negatively correlated with family conflict (r = -0.03, P<0.05); whether parents were blocked from WeChat moments was significantly positively correlated with family function(r = -0.13, P<0.01), parental management (r = -0.14, P<0.01)

**Table 2. Differences in attitudes towards adding parents as WeChat friends among college students with different characteristics.**

| | | M±SD | T/F | P |
|---|---|---|---|---|
| Gender | Male | 4.32±0.82 | 2.99** | 0.003 |
| | Female | 4.25±0.78 | | |
| Grade | 1 | 4.32±0.76 | 12.343*** Year 2Year 1/3 | 0.000 |
| | 2 | 4.22±0.83 | | |
| | 3 | 4.34±0.77 | | |
| Major | Liberal Arts | 4.30±0.78 | 8.002*** Fine Arts<Liberal Arts/Science/Physical Education | 0.000 |
| | Science | 4.25±0.81 | | |
| | Fine Arts | 4.15±0.86 | | |
| | Physical Education | 4.38±0.78 | | |
| Hometown | Rural | 4.28±0.80 | 0.515 | 0.615 |
| | Urban | 4.26±0.80 | | |
| Father Education Level | ≤High School | 4.26±0.80 | -3.73*** | 0.000 |
| | ≥College | 4.41±0.76 | | |
| Mother Education Level | ≤High School | 4.27±0.80 | -2.39* | 0.017 |
| | ≥College | 4.37±0.80 | | |
| Parents' Marital Status | Good | 4.36±0.76 | 18.24*** | 0.000 |
| | Poor | 3.88±0.87 | | |

Note:

* indicates P < 0.05

** indicates P < 0.01

*** indicates P < 0.001.

was significantly negatively correlated and significantly positively correlated with family conflict (r = -0.10, P<0.01).

Attitudes toward parental WeChat friend requests were significantly positively correlated with family function satisfaction (r = 0.44, P<0.01), parental management (r = 0.34, P<0.01), and significantly negatively correlated with family conflict (r = -0.33, P<0.01).

### Regression analysis of factors influencing responses to parental WeChat friend requests and moments blocking

The variables grade (1 = first grade, 2 = second grade, 3 = third grade), father's education (1 = high school and below, 2 = specialist and above), parent's marriage (1 = poor, 2 = good), family function, parental management, and family conflict that were statistically significant in the differences in the proportion of parents' WeChat friends requests and the correlation analysis were taken as the independent variables, and the situation of the parents' WeChat friends (0 = No, 1 = Yes) was taken as the dependent variable to perform a multifactor logistic regression analysis. The results found that parental management (OR = 1.044, 95% CI = 1.029~1.060) was an influential factor for the response of college students add their parents as WeChat friends, controlling for grade level and parents' marital status, and college students with good parental management were more inclined to parental WeChat friend requests. See Table 3.

Differences in the proportion of blocking parents from viewing WeChat moments and the variables gender (1 = male, 2 = female), major (1 = Arts, 2 = Science, 3 = Arts, 4 = Physical Education), parents' marital status (1 = bad, 2 = good), family function, parental management, and family conflict that yielded statistical significance based on correlation analyses were used

**Table 3. Logistic regression analysis of factors influencing responses of parents WeChat as friend requests.**

| Independent Variable | Reference Group | β | SE | Waldχ² Value | P Value | OR Value | 95% CI |
|---|---|---|---|---|---|---|---|
| Grade | | | | | | | |
| 2 | 1 | -0.251 | 0.117 | 4.621 | 0.032 | 0.778 | 0.619~0.978 |
| 3 | | -0.182 | 0.161 | 1.288 | 0.257 | 0.833 | 0.608~1.142 |
| Father Education Level | | | | | | | |
| Collge Or Higher | High School or Lower | 0.350 | 0.221 | 2.508 | 0.113 | 1.419 | 0.920~2.189 |
| Parents Marital Status | | | | | | | |
| Good | Poor | 2.040 | 0.105 | 374.566 | 0.000 | 7.691 | 6.256~9.457 |
| Family Function | | 0.029 | 0.021 | 1.924 | 0.165 | 1.029 | 0.988~1.072 |
| Parental Management | | 0.043 | 0.008 | 31.955 | 0.000 | 1.044 | 1.029~1.060 |
| Family Conflict | | 0.031 | 0.022 | 2.075 | 0.150 | 1.032 | 0.989~1.077 |

Note: Family Function, Parental Management, and Family Conflict are continuous variables, the same below.

as the independent variables, and blocking WeChat moments (0 = No, 1 = Yes) was used as the dependent variable in a multi-factor Logistic regression analysis. The results found that family function degree (OR = 0.915, 95% CI = 0.882~0.949), parental management (OR = 0.937, 95% CI = 0.924~0.950), and family conflict (OR = 1.101, 95% CI = 1.060~1.143) were the influencing factors for college students' blocking their parents from viewing their WeChat moments, controlling for gender. College students with poor family function, poor parental management, and serious family conflict are more inclined to block their parents' from viewing their WeChat moments. See Table 4.

## Regression analysis of the attitudes towards parental WeChat friends requests

The statistically significant variables in the analysis of variance and correlation: gender, grade, major, father's education, mother's education, parent's marriage, family function, parental management, and family conflict were used as independent variables, and the college students' willingness towards their parents becoming WeChat friend requests were used as the dependent variable to conduct stepwise linear hierarchical regression analyses. In the first step, demographic variables were included to fit a multiple linear regression model of college

**Table 4. Logistic regression analysis of factors influencing college students' blocking parents on moments.**

| Independent Variable | Reference Group | β | SE | Waldχ² Value | P Value | OR Value | 95% CI |
|---|---|---|---|---|---|---|---|
| Gender | | | | | | | |
| Female | 男 | 0.596 | 0.106 | 31.347 | 0.000 | 1.814 | 1.473~2.235 |
| Major | | | | | | | |
| Science | Liberal Arts | -0.172 | 0.109 | 2.491 | 0.115 | 0.842 | 0.680~1.043 |
| Fine Arts | | 0.111 | 0.152 | 0.527 | 0.468 | 1.117 | 0.829~1.506 |
| Physical Science | | -0.352 | 0.247 | 2.034 | 0.154 | 0.703 | 0.433~1.141 |
| Parents' Marital Status | | | | | | | |
| Good | Poor | -0.028 | 0.111 | 0.064 | 0.800 | 0.972 | 0.782~1.209 |
| Family Function | | -0.089 | 0.019 | 22.894 | 0.000 | 0.915 | 0.882~0.949 |
| Parental Managemet | | -0.065 | 0.007 | 88.235 | 0.000 | 0.937 | 0.924~0.950 |
| Family Conflict | | 0.096 | 0.019 | 25.062 | 0.000 | 1.101 | 1.060~1.143 |

**Table 5. Regression analysis of college students' response to parental WeChat friend requests.**

| Variable | Model 1 | | | Model 2 | | |
|---|---|---|---|---|---|---|
| | β Value | T Value | P Value | β Value | T Value | P Value |
| Gender | -0.055 | -2.637 | 0.000 | -0.043 | -2.307 | 0.021 |
| Grade | -0.001 | -0.037 | 0.970 | 0.034 | 2.649 | 0.008 |
| Major | -0.027 | -2.231 | 0.026 | -0.019 | -1.780 | 0.075 |
| Father Education Level | 0.104 | 2.459 | 0.014 | 0.016 | 0.436 | 0.663 |
| Mother Education Level | 0.057 | 1.196 | 0.232 | 0.006 | 0.139 | 0.890 |
| Parents' Marital Status | 0.478 | 18.132 | 0.000 | 0.273 | 11.508 | 0.000 |
| Family Function | - | - | - | 0.082 | 22.331 | 0.000 |
| Parental Management | - | - | - | 0.021 | 15.592 | 0.000 |
| Family Conflict | - | - | - | -0.056 | -13.712 | 0.000 |
| R2 | 0.055 | | | 0.271 | | |
| Δ R2 | 0.054 | | | 0.270 | | |
| F | 59.641*** | | | 254.616 | | |

students' attitudes toward parental WeChat friends. The results showed that gender (female) and major (fine arts) had a negative predictive effect on attitudes toward parental WeChat friends (= -0.055, -0.027, both P-values <0.05), and that fathers' educational level (specialist and above) and parents' marriage (good) had a positive predictive effect (= 0.104, 0.478, P-value all <0.05). In the second step, we further explored the effects of family function, parental management, and family conflict on parental WeChat friend requests, and found that gender, grade, parental marriage, family function, parental management, and family conflict jointly predicted 27.0% of the variance in attitudes about parents becoming WeChat friends; it can be seen that family function, parental management, and family conflict are influences on attitudes about adding parents as WeChat friends, (= 22.331, 15.592, P<0.001), and family conflict negatively predicted response towards adding parents' as WeChat friends (= -0.056, P<0.001).See Table 5.

## Discussion

According to the results, 92.5% of college students have all their parents as WeChat friends, only 9.4% of students chose to block their parents on WeChat Moments, and college students' feeling scores of having their parents as their WeChat friends are higher than the median value, This validates Research Hypothesis 1 (H1): Chinese college students maintain a positive WeChat parent-child relationship.

These results indicate that college students' overall attitudes towards their parents WeChat friend requests are positive, reflecting that in the age of social media that using WeChat to maintain relationship links with parents is regarded as a common and effective means by college students. The option of blocking the content of Moments after being added as a friend is seen by CMP as one of the individual's responses to facing privacy turbulence [3]. After being blocked, parents still use WeChat as a tool for communication with their children, but it is regarded as the same as being explicitly told by the children that their parents are unable to share the content of their Moments as co-owners, for Chinese society, where filial piety is the cultural foundation, this is tantamount to a head-on conflict with parents. Even in modern Chinese society, where young people endeavour to pursue their "selves", avoiding conflict and pursuing "peace is precious" remain the core norms of the Chinese family and society.

However, at the same time, college students, being skilled users of social media, may have fully assessed the possible privacy risks associated with their parents adding them as WeChat

friends and adapted their communication privacy management strategies of WeChat before adding them as friends [4], For example, special WeChat friend grouping settings for parents (to ensure that the content of the Moments was only open to specific groups), or deleted the content of Moments that they considered unsuitable for their parents to view, in order to ensure positive impression management as accurately as possible through the setting of self-disclosure boundaries.

Role theory in social psychology proposes that individuals in society play different roles (e.g., children, classmates, colleagues, etc.), and that different roles correspond to different role expectations, and that people will adjust their impression management styles according to the differences in expectations in order to obtain positive evaluations [13, 36, 37]. College students can set self-disclosure boundaries by clarifying that the content posted in Moments corresponds to the correct role expectations through boundary-setting means such as grouping (this approach to privacy management is more euphemistic and less likely to cause conflict than outright blocking),which may be one of the reasons for their positive attitudes towards adding their parents as WeChat friends.

Students with highly educated parents have a more positive attitude towards adding their parents as WeChat friends, probably because parents with higher education are more likely to accept modern family education concepts, and are more likely to master the scientific way of communicating with their children, which is relatively better for parent-child interaction. Parents' good marital relationship has a positive influence on college students' adding their parents as friends, while children in families with poor marital relationship are more likely to block their parents, which confirms the relevance of parents' marital status to children's self-disclosure willingness,Golish and Caughlin also confirmed this correlation by examining differences in self-disclosure intentions between children from divorced and non-divorced families [38]. Overall, in families with good parent-child relationships (characterized by high levels of family function and effective parental management), children are more likely to accept their parents' WeChat friend requests, holding a more positive attitude towards adding parents as friends and reducing the inclination to block them. Conversely, in families with poor parent-child relationships (marked by low levels of family function satisfaction and ineffective parental management), children tend to display more negative attitudes towards adding parents as WeChat friends and are more likely to resort to blocking their parents as a means of managing privacy on WeChat. This supports Research Hypotheses 2 and 3. Specifically, Hypothesis 2 (H2), which posits that family function satisfaction positively influences the WeChat parent-child relationship among college students, and Hypothesis 3 (H3), which suggests that parental management also has a positive influence on this relationship, are both validated by the findings.

For families, equal, harmonious and trusting parent-child relationships are also built on scientific parental management tools. We investigated parental management before children enter university, because most children live with their parents before they enter university, which is an important time for establishing emotional attachment between parents and children, and the results show that if parents at this time lack understanding of their children's daily situation and lack of parental companionship, they will be categorized as families with poor parental management, which is not easy to establish a good parent-child attachment relationship, and this will lead to college students from such families showing more negative attitudes towards adding their parents as WeChat friends. When we examine the level of parental management, the items mainly includes the degree of parental knowledge of their children's situation (children's social relations, daily activity trajectory, etc.), parents with a high degree of knowledge of their children's situation,indicating higher scores in parental management, may imply that their family privacy orientation involves children having greater transparency

with their parents and engaging in more frequent daily communication. This could be reflected in their WeChat parent-child relationships: children have a more positive attitude towards adding their parents as WeChat friends, and children are less likely to block their parents on WeChat moments. Therefore, in the context of this study, a high level of parental management implies good family communication skills, which is also positively related to family cohesion [39]. Families with high scores in family function imply a more democratic, equal, and harmonious parent-child relationship by testing the Adaptation, Partnership, Growth, Affection, and Resolve among family members. This contributes to enhancing the quality of parent-child relationships and the establishment of trust between parents and children. The positive impact of this on children adding their parents as social media friends has been validated [4].

On the contrary, the traditional parental management of absolute authority, although to some extent it can be mandatory to ensure that children's privacy is shared, can be seen as a harm to the child's sense of "self" and an invasion of privacy,especially as the child moves away from the family to become independent,and the child will become resentful and rebellious towards the parental management of this kind.Therefore, good parental management should avoids the unequal family power structure, learns to respect the children's sense of self and privacy boundaries, and creates a good atmosphere of parent-child communication, otherwise, even if the children add their parents to WeChat as their friends on the basis of "avoidance of conflict", they will also use blocking and other means to avoid the parents' "prying" into their self-disclosure.

Constant conflict and arguments within a family can greatly disrupt a harmonious and friendly family atmosphere, making it difficult to establish a conducive environment for parent-child communication and the development of intimate relationships. Children who experience prolonged exposure to negative emotions within the family may develop a diminished sense of familial affection and suffer psychological harm. They may even develop a desire to escape the negative influences of the family environment. This is why children from families with serious conflicts are more reluctant to add their parents as WeChat friends and are more inclined to block their parents on WeChat moments compared to children from harmonious family environments. This confirms Research Hypothesis 4 (H4), which posits that family conflict negatively influences the WeChat parent-child relationship among college students.

## Conclusion, limitation and future research

In summary, this study provides an in-depth exploration of Chinese college students' response and attitudes toward their parents' requests to be added as WeChat friends, and discusses factors such as traditional culture, as well as the quality of parent-child relationships and family environment in the context of new media. This study used convenience sampling to distribute questionnaires to more than 6,000 college students from a university in Chengdu, covering a number of factors, including grade, gender, major, parental education, marital relationship, parental management, family caring degree and family conflict degree, etc., in order to gain a comprehensive understanding of the college students' response toward their parents' WeChat friend requests. The results showed that the majority of college students were generally receptive to their parents' friend requests, and only a small number of students chose to block their parents on WeChat moments, which reflects the important role that social media plays in parent-child relationships; At the same time, we also believe that this kind of WeChat friendship and "non-blocking" with parents, in addition to using WeChat as a tool to maintain a relationship with parents, may also be based on "filial piety", "harmony is precious" and other Chinese family and social concepts, so the acceptance of parental friend requests and non-blocking of

moments does not mean that college students will not use more subtle means to draw boundaries with their parents on the degree of self-disclosure.

College students' attitudes towards adding their parents as WeChat friends vary according to grade, gender, and major, but more importantly, their parents' marital relationship, parent-child relationship, and family environment are also important factors influencing college students' response towards adding their parents as WeChat friends. Good husband-wife relationship, scientific parental management, good family function and a more harmonious family atmosphere all enhance the positive attitudes of college students towards adding their parents as WeChat friends, which reveals that in the context of new media, the family structure in China has been transformed from one-way authority to two-way authority, which requires parents to establish modern and scientific parenting concepts, and to create a harmonious and loving family atmosphere full of trust and mutual respect for their children, which is conducive to the establishment of a good parent-child relationship, and directly affects their children's desire to communicate and share with their parents, which will show intuitive differences in the construction of parent-child relationships on social media platforms.

The limitations of this study primarily include: first,the lack of further investigation into methods of personal boundary setting beyond blocking, such as the grouping method, the type of content visible and invisible to parents, etc., which can further improve the correlation between college students' self-disclosure mechanism and parent-child relationship; second, the examination of influencing factors can be further incorporated into the examination of the college students' personalities, communication privacy management habits, social media use habits and other factors, and the distinction between fathers and mothers can be examined. Third, this study employed convenience sampling, which may limit the generalizability of the findings. Future research will expand the sampling scope by selecting participants from representative cities across different economic levels and regions in China. Additionally, future studies will incorporate factors such as personality traits, levels of privacy management, and social media dependency to develop a more comprehensive and robust research framework.

## Author Contributions

**Conceptualization:** Zhao Feng, Chun-Mei Hu.

**Data curation:** Chun-Mei Hu.

**Investigation:** Zhao Feng, Ling-Ling He.

**Methodology:** Chun-Mei Hu.

**Resources:** Ling-Ling He.

**Writing – original draft:** Zhao Feng.

**Writing – review & editing:** Zhao Feng.

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
