## [Decision Letter · Decision Letter 0]

28 Aug 2024

PONE-D-24-28753Boundaries with Parents: A Study of Chinese College Students' Responses to Parental WeChat Friend Requests and Its Implications on Parent-Child RelationshipsPLOS ONE

Dear Dr. Hu,

Thank you for submitting your manuscript to PLOS ONE. After careful consideration, we feel that it has merit but does not fully meet PLOS ONE’s publication criteria as it currently stands. Therefore, we invite you to submit a revised version of the manuscript that addresses the points raised during the review process.

Your paper needs a revision under the light of reviewers' comments ad suggestions.==============================

We look forward to receiving your revised manuscript.

Kind regards,

Farooq Ahmed, PhD

Academic Editor

PLOS ONE

Journal Requirements:

Reviewers' comments:

Reviewer's Responses to Questions

**Comments to the Author**

1. Is the manuscript technically sound, and do the data support the conclusions?

Reviewer #1: Yes

Reviewer #2: Yes

2. Has the statistical analysis been performed appropriately and rigorously? 

Reviewer #1: Yes

Reviewer #2: Yes

3. Have the authors made all data underlying the findings in their manuscript fully available?

Reviewer #1: Yes

Reviewer #2: Yes

4. Is the manuscript presented in an intelligible fashion and written in standard English?

Reviewer #1: Yes

Reviewer #2: Yes

5. Review Comments to the Author

Reviewer #1: The paper was well-written and included rigorous statistical analysis. Authors highlighted the critical matter of the digital world concerning the parent-child relationship. Upon reviewing this paper, I recommend incorporating the following changes by the scholar/author/researcher. These changes have also been highlighted separately in the paper. The paper has the potential for publication, considering the incorporation of significant changes.

1. Your study represents the responses of only one small area of China, i.e., Chengdu. While selecting only one university, it cannot be generalized to the whole of China as the title mention. You can justify it in methods.

2. A minor proofreading is required (mistakes are mentioned in attached paper).

3. An empirical research needs a hypothesis. I would suggest adding hypotheses in your introduction.

4. References should be in Vancouver style. Follow the guidelines of the Journal.

Reviewer #2: 1) The manuscript mentions that all participants were over 18 years of age but fails to provide a more detailed age range for the participants. I recommend that the authors provide a clear and specific age range for the participants in the manuscript in abstract, method section (under heading of participant’s selection) and results. This should include the minimum and maximum ages and the mean or median age if possible. Additionally, discussing the distribution of participants across different age groups just like other demographics would offer valuable context for interpreting the findings. Including this information will enhance the rigor of the study and provide a more nuanced understanding of the results, allowing for better comparisons with existing literature.

2) I recommend the authors explicitly identify and articulate the specific type (s) of research gap(s) in the literature review. This will clarify the study's contribution and enhance its scholarly impact and relevance.

3) The manuscript does not explicitly describe the research design, making it difficult to understand the overall approach taken to conduct the study. Without this clarity, it is challenging to assess the appropriateness of the methods used and the validity of the findings.

4) The insufficient theoretical and conceptual framework diminishes the clarity of the research. Incorporating relevant, well-established theories is essential to strengthen the study's foundation and academic significance

5) While the research questions are provided, the absence of clearly defined research objectives and hypotheses is a significant oversight. Without these, the study lacks direction and focus, making it difficult to assess the alignment of the research with its theoretical framework and identified gap. I strongly recommend formulating specific objectives and hypotheses to guide the study’s methodology and analysis, ensuring a more coherent and robust research design.

6) Explain how the sampling method (e.g., convenience sampling) aligns with the research design and objectives. The use of convenience sampling in this study introduces potential biases and limits generalizability, as the sample may not represent the broader population of college students (collected through one university of a specific region i.e. Chengdu). Alternative methods like stratified random or cluster sampling would enhance validity. If convenience sampling is necessary, the authors should acknowledge its limitations and discuss how they may affect the study's findings. It is important to provide a clear rationale for this choice in the manuscript.

7) The manuscript inadequately explains how the analytical methods specifically address the research questions (RQ1, RQ2, & RQ3) leading to uncertainty about the study's validity. Further, discussion section also lacks the effective validation of these three research questions introduced earlier, leading to a disconnect between the proposed questions and findings of study. It is suggested to integrate and address these questions in the discussion to ensure aligned and well-supported conclusions.

6. PLOS authors have the option to publish the peer review history of their article (what does this mean?). If published, this will include your full peer review and any attached files.

Reviewer #1: **Yes: **Shagufta Hamid Ali

Reviewer #2: **Yes: **Prof. Dr. Muhammad Saleem

---

## [Author Response · Author response to Decision Letter 0]

11 Sep 2024

Thank you to the reviewers for their valuable feedback. We have carefully reviewed and considered the comments, and have revised the paper accordingly. Below are our responses to each comment.

Reviewer #1: The paper was well-written and included rigorous statistical analysis. Authors highlighted the critical matter of the digital world concerning the parent-child relationship. Upon reviewing this paper, I recommend incorporating the following changes by the scholar/author/researcher. These changes have also been highlighted separately in the paper. The paper has the potential for publication, considering the incorporation of significant changes.

1.Your study represents the responses of only one small area of China, i.e., Chengdu. While selecting only one university, it cannot be generalized to the whole of China as the title mention. You can justify it in methods.

Authors Response:

There is a clerical error here, and we actually covered more than just one colleges in Chengdu.We have made revision in methods as：Convenience sampling was employed to select 6159 freshmen to junior students to participate in the questionnaire survey at colleges in Chengdu, which is a representative core city in western China. Chengdu, as the central city in western China, attracts students from across the country, making its colleges a microcosm of diverse regional backgrounds. Therefore, we considered Chengdu a representative research location for exploring the attitudes and behaviors of Chinese college students. 

2.A minor proofreading is required (mistakes are mentioned in attached paper).

Authors Response:

The corrections have been made in response to the annotations in the attachment.

3.An empirical research needs a hypothesis. I would suggest adding hypotheses in your introduction.

Authors Response:

4 hypotheses have been added in introduction part:

Hypothesis 1 (H1): 

Chinese college students maintain a positive WeChat parent-child relationship.

Hypothesis 2 (H2): Family function satisfaction will positively influence college students' WeChat parent-child relationships.

Hypothesis 3 (H3): Parental management will positively influence college students' WeChat parent-child relationships.

Hypothesis 4 (H4): Family conflict will negatively influence college students' WeChat parent-child relationships.

4. References should be in Vancouver style. Follow the guidelines of the Journal.

Authors Response:

References have been changed to Vancouver style.

Reviewer #2: 

1)The manuscript mentions that all participants were over 18 years of age but fails to provide a more detailed age range for the participants. I recommend that the authors provide a clear and specific age range for the participants in the manuscript in abstract, method section (under heading of participant’s selection) and results. This should include the minimum and maximum ages and the mean or median age if possible. Additionally, discussing the distribution of participants across different age groups just like other demographics would offer valuable context for interpreting the findings. Including this information will enhance the rigor of the study and provide a more nuanced understanding of the results, allowing for better comparisons with existing literature.

Authors Response:

we have provided a more detailed age range for the participants in the manuscript:The age range is 18-25 years. Specifically, there are 2,143 individuals aged 18 (34.8%), 2,459 individuals aged 19 (39.9%), and 1,557 individuals aged 20-25 (25.3%) (since the number of individuals aged 21-25 is relatively small, they have been combined into the 20-25 age group). These changes have been reflected in the abstract, method section (under the heading of participant selection), and results(table 2) as suggested.

2) I recommend the authors explicitly identify and articulate the specific type (s) of research gap(s) in the literature review. This will clarify the study's contribution and enhance its scholarly impact and relevance.

Authors Response:

we have clarified the specific research gap: currently, there is a scarcity of quantitative research focusing on the concept of WeChat-based parent-child relationships. Our study aims to address this gap by providing empirical data and analysis in this under-explored area. We believe that this contribution will enhance the scholarly understanding of parent-child dynamics in the context of new media platforms like WeChat.

2)The manuscript does not explicitly describe the research design, making it difficult to understand the overall approach taken to conduct the study. Without this clarity, it is challenging to assess the appropriateness of the methods used and the validity of the findings.

Authors Response:

We have added research design in Method section:This study employs a cross-sectional design. A questionnaire survey is used, which consists of two parts. The first part covers demographic variables, including age, gender, grade level, hometown, major, father's education level, mother's education level, and parental marital status. The second part includes the survey instruments: a questionnaire on response and attitudes towards parents' WeChat friend requests, a family function satisfaction scale, a parental management scale, and a family conflict scale. Chi-square test was employed to assess the relationship between categorical variables

3)The insufficient theoretical and conceptual framework diminishes the clarity of the research. Incorporating relevant, well-established theories is essential to strengthen the study's foundation and academic significance

Authors Response:

We applied the theoretical framework of Communication Privacy Management (CPM), specifically focusing on the concepts of privacy ownership, privacy control, and privacy turbulence, to explain college students' responses and attitudes toward their parents' WeChat friend requests, as well as their decisions to block or not block their parents from viewing their WeChat Moments. Based on this framework, we developed four research hypotheses.

5) While the research questions are provided, the absence of clearly defined research objectives and hypotheses is a significant oversight. Without these, the study lacks direction and focus, making it difficult to assess the alignment of the research with its theoretical framework and identified gap. I strongly recommend formulating specific objectives and hypotheses to guide the study’s methodology and analysis, ensuring a more coherent and robust research design.

Authors Response:

4 hypotheses have been added in introduction part:

Hypothesis 1 (H1): 

Chinese college students maintain a positive WeChat parent-child relationship.

Hypothesis 2 (H2): Family function satisfaction will positively influence college students' WeChat parent-child relationships.

Hypothesis 3 (H3): Parental management will positively influence college students' WeChat parent-child relationships.

Hypothesis 4 (H4): Family conflict will negatively influence college students' WeChat parent-child relationships.

6) Explain how the sampling method (e.g., convenience sampling) aligns with the research design and objectives. The use of convenience sampling in this study introduces potential biases and limits generalizability, as the sample may not represent the broader population of college students (collected through one university of a specific region i.e. Chengdu). Alternative methods like stratified random or cluster sampling would enhance validity. If convenience sampling is necessary, the authors should acknowledge its limitations and discuss how they may affect the study's findings. It is important to provide a clear rationale for this choice in the manuscript.

Authors Response:

We have acknowledge the limitations of convenience sampling in the section of conclusion and limitations as :this study employed convenience sampling, which may limit the generalizability of the findings. Future research will expand the sampling scope by selecting participants from representative cities across different economic levels and regions in China. Additionally, future studies will incorporate factors such as personality traits, levels of privacy management, and social media dependency to develop a more comprehensive and robust research framework.

7) The manuscript inadequately explains how the analytical methods specifically address the research questions (RQ1, RQ2, & RQ3) leading to uncertainty about the study's validity. Further, discussion section also lacks the effective validation of these three research questions introduced earlier, leading to a disconnect between the proposed questions and findings of study. It is suggested to integrate and address these questions in the discussion to ensure aligned and well-supported conclusions.

Authors Response:

we have removed the original research questions (RQ1, RQ2, & RQ3) and replaced them with four research hypotheses. These hypotheses have been thoroughly discussed and validated in the discussion section, ensuring better alignment between the proposed ideas and the study’s findings.

---

## [Editor Report · Decision Letter 1]

13 Sep 2024

PONE-D-24-28753R1Digital Boundaries: A Study on WeChat Parent-Child Relationships Among  Chinese College Students and Their Association with Family FactorsPLOS ONE

Dear Dr. Hu,

Thank you for submitting your manuscript to PLOS ONE. After careful consideration, we feel that it has merit but does not fully meet PLOS ONE’s publication criteria as it currently stands. Therefore, we invite you to submit a revised version of the manuscript that addresses the points raised during the review process.

**Dear authors, Thanks you for revising the manuscript according to reviewers' comments. However, references are not according to journal requirement. Please revise the references according to guidelines.**

**
https://journals.plos.org/plosone/s/submission-guidelines#loc-references
**

We look forward to receiving your revised manuscript.

Kind regards,

Farooq Ahmed, PhD

Academic Editor

PLOS ONE
---

## [Author Response · Author response to Decision Letter 1]

18 Sep 2024

Thank you to the reviewers for their valuable feedback. We have carefully reviewed and considered the comments, and have revised the paper accordingly. Below are our responses to each comment.

Reviewer #1: The paper was well-written and included rigorous statistical analysis. Authors highlighted the critical matter of the digital world concerning the parent-child relationship. Upon reviewing this paper, I recommend incorporating the following changes by the scholar/author/researcher. These changes have also been highlighted separately in the paper. The paper has the potential for publication, considering the incorporation of significant changes.

1.Your study represents the responses of only one small area of China, i.e., Chengdu. While selecting only one university, it cannot be generalized to the whole of China as the title mention. You can justify it in methods.

Authors Response:

There is a clerical error here, and we actually covered more than just one colleges in Chengdu.We have made revision in methods as：Convenience sampling was employed to select 6159 freshmen to junior students to participate in the questionnaire survey at colleges in Chengdu, which is a representative core city in western China. Chengdu, as the central city in western China, attracts students from across the country, making its colleges a microcosm of diverse regional backgrounds. Therefore, we considered Chengdu a representative research location for exploring the attitudes and behaviors of Chinese college students. 

2.A minor proofreading is required (mistakes are mentioned in attached paper).

Authors Response:

The corrections have been made in response to the annotations in the attachment.

3.An empirical research needs a hypothesis. I would suggest adding hypotheses in your introduction.

Authors Response:

4 hypotheses have been added in introduction part:

Hypothesis 1 (H1): 

Chinese college students maintain a positive WeChat parent-child relationship.

Hypothesis 2 (H2): Family function satisfaction will positively influence college students' WeChat parent-child relationships.

Hypothesis 3 (H3): Parental management will positively influence college students' WeChat parent-child relationships.

Hypothesis 4 (H4): Family conflict will negatively influence college students' WeChat parent-child relationships.

4. References should be in Vancouver style. Follow the guidelines of the Journal.

Authors Response:

References have been changed to Vancouver style.

The References have been revised by following the guidelines of the journal.

Reviewer #2: 

1)The manuscript mentions that all participants were over 18 years of age but fails to provide a more detailed age range for the participants. I recommend that the authors provide a clear and specific age range for the participants in the manuscript in abstract, method section (under heading of participant’s selection) and results. This should include the minimum and maximum ages and the mean or median age if possible. Additionally, discussing the distribution of participants across different age groups just like other demographics would offer valuable context for interpreting the findings. Including this information will enhance the rigor of the study and provide a more nuanced understanding of the results, allowing for better comparisons with existing literature.

Authors Response:

we have provided a more detailed age range for the participants in the manuscript:The age range is 18-25 years. Specifically, there are 2,143 individuals aged 18 (34.8%), 2,459 individuals aged 19 (39.9%), and 1,557 individuals aged 20-25 (25.3%) (since the number of individuals aged 21-25 is relatively small, they have been combined into the 20-25 age group). These changes have been reflected in the abstract, method section (under the heading of participant selection), and results(table 2) as suggested.

2) I recommend the authors explicitly identify and articulate the specific type (s) of research gap(s) in the literature review. This will clarify the study's contribution and enhance its scholarly impact and relevance.

Authors Response:

we have clarified the specific research gap: currently, there is a scarcity of quantitative research focusing on the concept of WeChat-based parent-child relationships. Our study aims to address this gap by providing empirical data and analysis in this under-explored area. We believe that this contribution will enhance the scholarly understanding of parent-child dynamics in the context of new media platforms like WeChat.

2)The manuscript does not explicitly describe the research design, making it difficult to understand the overall approach taken to conduct the study. Without this clarity, it is challenging to assess the appropriateness of the methods used and the validity of the findings.

Authors Response:

We have added research design in Method section:This study employs a cross-sectional design. A questionnaire survey is used, which consists of two parts. The first part covers demographic variables, including age, gender, grade level, hometown, major, father's education level, mother's education level, and parental marital status. The second part includes the survey instruments: a questionnaire on response and attitudes towards parents' WeChat friend requests, a family function satisfaction scale, a parental management scale, and a family conflict scale. Chi-square test was employed to assess the relationship between categorical variables

3)The insufficient theoretical and conceptual framework diminishes the clarity of the research. Incorporating relevant, well-established theories is essential to strengthen the study's foundation and academic significance

Authors Response:

We applied the theoretical framework of Communication Privacy Management (CPM), specifically focusing on the concepts of privacy ownership, privacy control, and privacy turbulence, to explain college students' responses and attitudes toward their parents' WeChat friend requests, as well as their decisions to block or not block their parents from viewing their WeChat Moments. Based on this framework, we developed four research hypotheses.

5) While the research questions are provided, the absence of clearly defined research objectives and hypotheses is a significant oversight. Without these, the study lacks direction and focus, making it difficult to assess the alignment of the research with its theoretical framework and identified gap. I strongly recommend formulating specific objectives and hypotheses to guide the study’s methodology and analysis, ensuring a more coherent and robust research design.

Authors Response:

4 hypotheses have been added in introduction part:

Hypothesis 1 (H1): 

Chinese college students maintain a positive WeChat parent-child relationship.

Hypothesis 2 (H2): Family function satisfaction will positively influence college students' WeChat parent-child relationships.

Hypothesis 3 (H3): Parental management will positively influence college students' WeChat parent-child relationships.

Hypothesis 4 (H4): Family conflict will negatively influence college students' WeChat parent-child relationships.

6) Explain how the sampling method (e.g., convenience sampling) aligns with the research design and objectives. The use of convenience sampling in this study introduces potential biases and limits generalizability, as the sample may not represent the broader population of college students (collected through one university of a specific region i.e. Chengdu). Alternative methods like stratified random or cluster sampling would enhance validity. If convenience sampling is necessary, the authors should acknowledge its limitations and discuss how they may affect the study's findings. It is important to provide a clear rationale for this choice in the manuscript.

Authors Response:

We have acknowledge the limitations of convenience sampling in the section of conclusion and limitations as :this study employed convenience sampling, which may limit the generalizability of the findings. Future research will expand the sampling scope by selecting participants from representative cities across different economic levels and regions in China. Additionally, future studies will incorporate factors such as personality traits, levels of privacy management, and social media dependency to develop a more comprehensive and robust research framework.

7) The manuscript inadequately explains how the analytical methods specifically address the research questions (RQ1, RQ2, & RQ3) leading to uncertainty about the study's validity. Further, discussion section also lacks the effective validation of these three research questions introduced earlier, leading to a disconnect between the proposed questions and findings of study. It is suggested to integrate and address these questions in the discussion to ensure aligned and well-supported conclusions.

Authors Response:

we have removed the original research questions (RQ1, RQ2, & RQ3) and replaced them with four research hypotheses. These hypotheses have been thoroughly discussed and validated in the discussion section, ensuring better alignment between the proposed ideas and the study’s findings.

---

## [Editor Report · Decision Letter 2]

20 Sep 2024

Digital Boundaries: A Study on WeChat Parent-Child Relationships Among  Chinese College Students and Their Association with Family Factors

PONE-D-24-28753R2

Dear Dr.Chun-mei Hu,

We’re pleased to inform you that your manuscript has been judged scientifically suitable for publication and will be formally accepted for publication once it meets all outstanding technical requirements.

Kind regards,

Farooq Ahmed, PhD

Academic Editor

PLOS ONE

---

## [Editor Report · Acceptance letter]

16 Oct 2024

PONE-D-24-28753R2 

PLOS ONE

Dear Dr. Hu, 

I'm pleased to inform you that your manuscript has been deemed suitable for publication in PLOS ONE. Congratulations! Your manuscript is now being handed over to our production team.

Kind regards, 

on behalf of

Dr. Farooq Ahmed 

Academic Editor

PLOS ONE